**Subject Category:**
Biology (whole organism)

biomechanics/ecology/evolution

force measurements, radula, feeding, functional morphology, nanoindentation

**Author for correspondence:**
Wencke Krings
e-mail: wencke.krings@uni-hamburg.de

Electronic supplementary material is available at https://dx.doi.org/10.6084/m9.figshare.c.4540142.

# In slow motion: radula motion pattern and forces exerted to the substrate in the land snail *Cornu aspersum* (Mollusca, Gastropoda) during feeding

Wencke Krings[1], Taissa Faust[1], Alexander Kovalev[2], Marco Thomas Neiber[1], Matthias Glaubrecht[1] and Stanislav Gorb[2]

[1]Center of Natural History (CeNak), University of Hamburg, Martin-Luther-King-Platz 3, 20146 Hamburg, Germany
[2]Functional Morphology and Biomechanics, Zoological Institute of the University of Kiel, Am Botanischen Garten 9, 24118 Kiel, Germany

WK, 0000-0003-2158-9806; AK, 0000-0002-9441-5316; SG, 0000-0001-9712-7953

The radula is the anatomical structure used for feeding in most species of Mollusca. Previous studies have revealed that radulae can be adapted to the food or the substrate the food lies on, but the real, *in vivo* forces exerted by this organ on substrates and the stresses that are transmitted by the teeth are unknown. Here, we relate physical properties of the radular teeth of *Cornu aspersum* (Müller. 1774 *Vermium terrestrium et fluviatilium, seu animalium infusoriorum, helminthicorum, et testaceorum, non marinorum, succincta historia. Volumen alterum.* Heineck & Faber, Havniæ & Lipsiæ.), a large land snail, with experiments revealing their radula scratching force. The radula motion was recorded with high-speed video, and the contact area between tooth cusps and the substrate was calculated. Forces were measured in all directions; highest forces (106.91 mN) were exerted while scratching, second highest forces while pulling the radula upwards and pressing the food against its counter bearing, the jaw, because the main ingesta disaggregation takes place during those two processes. Nanoindentation revealed that the tooth hardness and elasticity in this species are comparable to wood. The teeth are softer than some of their ingesta, but since the small

contact area of the tooth cusps (227 $\mu m^2$) transmits high local pressure (4698.7 bar) on the ingesta surface, harder material can still be cut or pierced with abrasion. This method measuring the forces produced by the radula during feeding could be used in further experiments on gastropods for better understanding functions and adaptations of radulae to ingesta or substrate, and hence, gastropods speciation and evolution.

## 1. Introduction

Mollusca are a species-rich animal group, second only to insects (e.g. [1]), containing around 80 000 recent species in the class Gastropoda alone [2]. In particular, the gastropods, snails and slugs, inhabit extraordinarily diverse environments: from marine and freshwater habitats to deserts, from hydrothermal vents in the deep sea to mountaintops, from rocky areas to rainforests and urban regions. The colonialization of diverse ecosystems goes along with the establishment of different ecological niches: gastropods feed on a range of different food sources with different mechanical properties; they can be herbivores, detritus feeders, predators, scavengers, parasites and ciliary feeders. This is possible because they carry a key innovation for mechanical food processing, the radula.

Naturalists of the nineteenth and early twentieth century equipped with simple light microscopes regarded the radula as an organ, which most Mollusca share (e.g. [3]), and later defined it as autapomorphy of this group. The first extensive study of this organ was done by Troschel [3], who introduced the feeding organ as the most important character complex for systematics at any level. Following this paradigm, gastropods were classified according to the 'form' of the radula and its teeth, replacing the more traditional view based on the use of shells only. Thiele [4] revised the Mollusca based on this new character, resulting in the reorganization of the systematics of gastropods at all taxonomic levels. Today, the systematic value of the radula morphology is generally accepted; however, its importance depends on the group. Some closely related species show extremely diverse radula morphologies (e.g. the Paludomidae from Lake Tanganyika; see [5]) and even specimens of the same species can produce differently shaped teeth, when fed with different foods [6,7]. As in other animals, processing ingesta mechanically, the morphology of the organ for food intake is nowadays understood as a link between the organisms and their environment, holding phylogenetic as well as ecological information.

In gastropods, this feeding organ, the buccal mass, includes not only the radula but also the jaw, the odontophore and numerous muscles. The odontophore is surrounded by the chitinous radula membrane [8], embedded with transverse and longitudinal rows of teeth that are sometimes mineralized. Radular teeth can show extremely different morphologies between taxa, but all possess a universal bauplan: base, stylus and cusp, the latter containing denticles (e.g. [9,10]). In general, radulae have been categorized by the number, type and arrangement of teeth (e.g. [11–14]). New radular teeth are formed continuously in the building zone of the radula ribbon before they enter the wearing zone (e.g. [15–22]). Only the anterior-most rows of teeth are used for feeding (e.g. [16–18,20–23]). The diversity of radulae and their teeth is not just defined by morphological differences in the tooth morphology, but there are also differences in material properties (e.g. [24]). To date, the vast majority of studies on radular teeth mineralization were conducted on the molluscan class Polyplacophora and on Patellogastropoda—a basal clade of Gastropoda. They gained interest because their tooth cusps are strongly wear resistant due to significant amounts of iron-based biominerals and silica incorporated in their chitinous teeth. Since the amounts of these minerals are different, the teeth have different mechanical properties, which make this feeding organ interesting for materials science (e.g. [25–38]). Additionally to Si, other elements like calcium and magnesium have been detected (e.g. [39–42]). However, physical properties of the vast majority of radular teeth, as well as its implications on their function, are still understudied.

While feeding, the muscles of the buccal mass protract and retract the odontophore [43] pulling the radula membrane across this cartilage (e.g. [14]). Even though the radula motion can be taxon-specific as well as food- and surface-specific (e.g. [44–46]), Mackenstedt & Märkel [47] described the overall feeding process for some terrestrial heterobranches as follows: in the resting position, the buccal mass is oriented horizontally and is not in contact with the mouth. Then, it is turned to a vertical position by muscles connecting the organ with the body wall. The radula lies on the odontophore and only the part outside of the sac is engaged in this process. The food is taken in by the movement of the radula over the tip of the odontophore and by the movement of the odontophore itself. Initially, the radula lies loose on the cartilage, but when it touches the substrate, the radula becomes pressed against the surface and thus comes in direct contact with the cartilage (a different relationship between radula

and odontophore has also been described by Morris & Hickman [48]: radula ribbon is tightly enrolled posteriorly and its anterior portion is protracted bringing teeth in the scratching position). Teeth come in contact with the food situated on the surface, and by scratching the surface gastropods effectively harvest particles of a variety of sizes. Feeding can be the grazing on soft substrate collecting microalgae but may also include cutting and grinding action, as potential ingesta may be too large to ingest at once or be fixed to the substrate. In those cases, the jaw, a reinforced part of the foregut cuticle, located opposite to the radula, may be used as counter bearing [49]. For example, a piece of a leaf can be squeezed between the jaw and radula and cut by the jaw.

Even though previous studies have revealed extensive information about the motion pattern and functioning of the radula and buccal mass, the feeding behaviour and radula kinetics (e.g. [44,48,50–62]) and even on the mechanical forces exerted by the radula [45,46,63], no research—to the best of our knowledge—has been done on the direct measurements of the mechanical forces of a living snail exerted on the surface by its feeding organ.

We used the Mediterranean land snail *Cornu aspersum* [64], previously known as *Helix aspersa*, belonging to the Gastropoda, Heterobranchia, Helicidae for this case study. It was only found at a few localities in Algeria [65] but is used for commercial breeding because of its size (body length = 11–12 cm). We chose it because of its large radula size, the availability and the easy handling. We recorded the detailed radula motion with a high-speed video camera and measured the exerted forces to the substrate while feeding and correlated this with the material properties of the teeth taking into account the contact area between tooth cusps and the substrate.

# 2. Material and methods

## 2.1. Animals and species assignment

Snails were obtained from Weinbergschneckenzucht Petra Sauer (Blankenburg, Germany). Animals (table 1) were kept at room temperature and all fed with the same organic vegetables (carrots, lettuce) and cuttlebones for two to three weeks prior to experiments. Specimens are inventoried in the Zoological Museum Hamburg (ZMH 150005).

To allocate the investigated specimens to one of the major mitochondrial lineages identified in *Cornu* [66] (see [66, p. 371] for the original description) by Guiller *et al*. [65], one specimen (table 1) was barcoded. Total genomic DNA was extracted using the protocol for degraded DNA described in Neiber *et al*. [67] from foot muscle tissue of a fully adult specimen. A fragment of the 16S rRNA gene was amplified using the primer pair 16Sar plus 16Sbr [68] and the polymerase chain reaction (PCR) protocol for the respective mitochondrial gene from Neiber & Hausdorf [69]. Both strands of the amplified product were sequenced at Macrogen Europe Laboratory (Amsterdam, The Netherlands) and assembled using ChromasPro 1.7.1 (Techelysium Pty Ltd, South Brisbane, Australia). The genetic identity of the examined specimen to specimens of the putative taxon referred to as *Helix aspersa maxima* [70] by Guiller *et al*. [65], to which the specimens correspond morphologically, was confirmed by a basis local alignment search tool (BLAST) search [71] in the GenBank database [72] resulting in a 99% best hit sequence identity to the partial sequence of the 16S rRNA gene catalogued under GenBank accession number AF126142 [65]. The newly generated sequence is deposited in GenBank (GenBank number MN080303).

It is important to note here that the status of the nominal taxon *Helix* (*Pomatia*) *aspersa* var. *maxima* [70] is unclear. Should further taxonomic research show that the taxon is indeed distinct from *Cornu aspersum* s. str., it cannot bear [70, p. 94] the name, which is nomenclatorially preoccupied by, e.g. *Helix nemoralis maxima* [73] (see [73, p. 7]). With the above said, we, therefore, refer the specimens studied here simply to *Cornu aspersum* [64] (see [64, p. 59] for the original description).

## 2.2. Radula, movement and contact area

Video imaging was performed on seven animals (five while feeding and on two without food; table 1) with a Nikon D810 (Nikkor lens AF-S VR Micro-Nikkor 105 mm f/2.8 G IF-ED) from the bottom and the side to detect the directions and the general motion pattern and chronological order of radula movement during feeding (see electronic supplementary material, Movie S1). The images were detailed enough to enable us to measure the area of the radula in contact with the substrate while feeding on five specimens (table 1 and figure 1*c*). Using graphical software Adobe® Photoshop® v. CS6, the area on different images was calculated resulting in an average area of contact.

**4**

**Table 1.** Snails used for different methods or experiments (barcoding, scanning electron microscope (SEM), nanoindentation, force measurements with $N$ of successful experiments and $N$ of force values, video imaging with radula-surface contact area measurement) with the individual weight in grams (average ± s.d.) measured before the experiment.

| snail number | weight (average ± s.d.) | used for: barcoding | SEM including tooth counting and tooth-area measurements | EDAX | force measurements: $N$ of successful experiments in each direction (and $N$ of measured force values) per snail | $N$ of days with experiments | videos | radula-surface contact area measurements | nanoindentation |
|---|---|---|---|---|---|---|---|---|---|
| ZMH 150005-1 | 20.68 ± 0.31 | | | | horizontal: 4 (5 anterior/3 posterior) vertical: 11 (28 up/17 down) | 4 | | | |
| ZMH 150005-2 | 18.55 ± 0.50 | | | | horizontal: 3 (16 anterior/5 posterior) vertical: 5 (13 up/9 down) | 2 | | | |
| ZMH 150005-3 | 20.64 ± 1.86 | | x | x | horizontal: 3 (6 anterior) vertical: 5 (16 up/13 down) | 2 | | | |
| ZMH 150005-4 | 22.16 | x | | | horizontal: 4 (16 anterior) | 1 | | | |
| ZMH 150005-6 | 23.40 | | x | x | horizontal: 5 (15 anterior/2 posterior) | 1 | | | |
| ZMH 150005-7 | 20.30 | | | | vertical: 7 (31 up/1 down) | 1 | x (with food) | x | x |
| ZMH 150005-8 | 18.52 ± 0.03 | | | | horizontal: 5 (6 anterior/1 posterior) vertical: 4 (9 up/5 down) | 2 | x (with food) | x | x |
| ZMH 150005-9 | 25.64 ± 3.14 | | x | x | horizontal: 5 (25 anterior/1 posterior) vertical: 4 (8 up/10 down) | 2 | | | |
| ZMH 150005-10 | 21.39 | | x | x | horizontal: 3 (12 anterior/4 posterior) | 1 | x (with food) | x | |
| ZMH 150005-11 | 16.94 | | | | horizontal: 5 (26 anterior/7 posterior) | 1 | | | |
| ZMH 150005-12 | 15.08 | | | | horizontal: 1 (2 anterior/1 posterior) | 1 | | | |
| ZMH 150005-13 | 22.81 | | | | horizontal: 2 (9 anterior/8 posterior) | 1 | | | |
| ZMH 150005-14 | 21.99 | | x | x | horizontal: 3 (5 anterior/4 posterior) vertical: 5 (10 up/12 down) | 1 | | | |
| ZMH 150005-15 | | | | | | | x (with food) | x | |
| ZMH 150005-16 | | | | | | | x (with food) | x | |
| ZMH 150005-17 | | | | | | | x (without food) | | |
| ZMH 150005-18 | | | | | | | x (without food) | | |

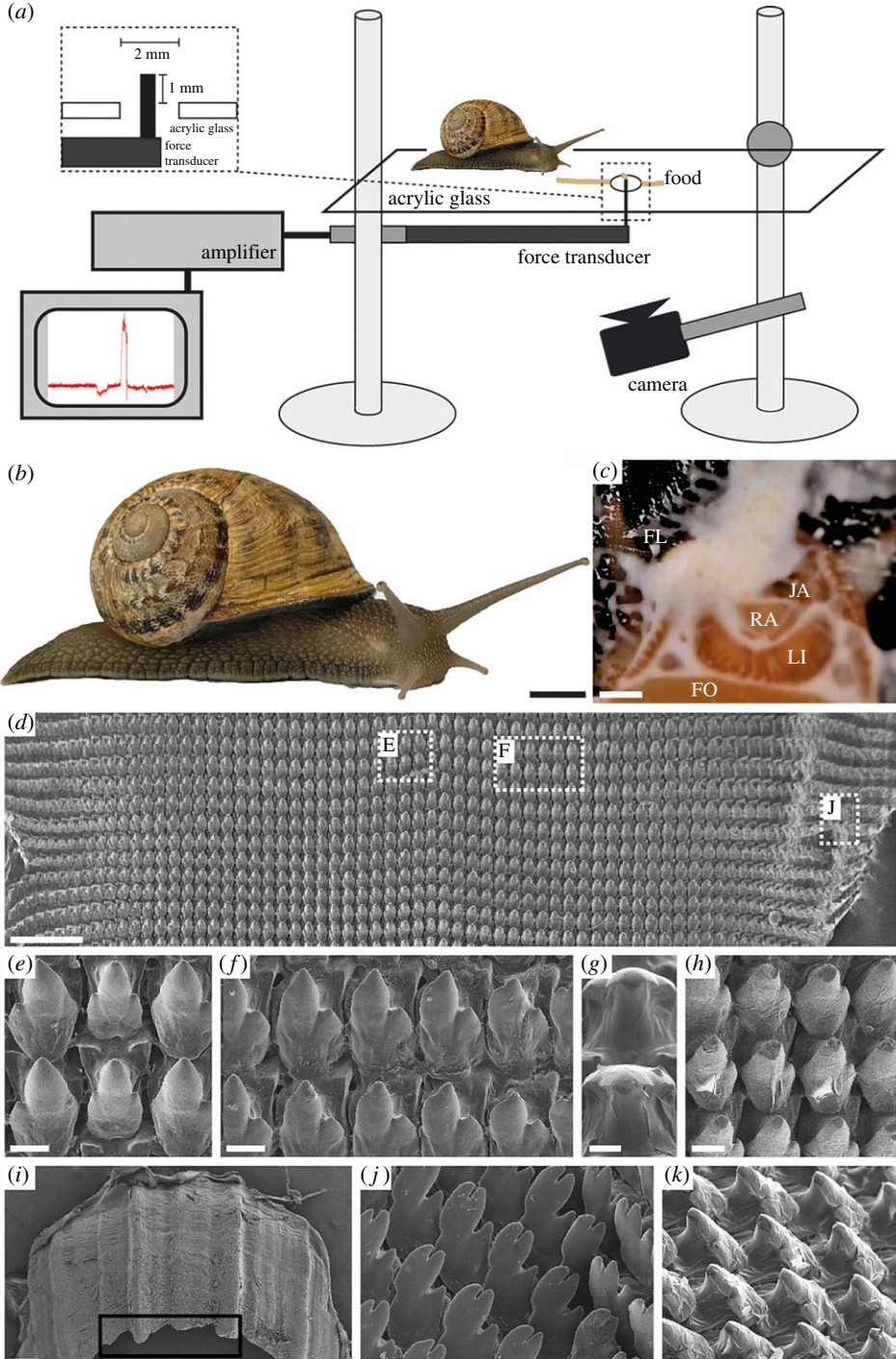

**Figure 1.** Experimental set-up, animals and images of the radular teeth. (*a*) The snails were put on a horizontal plastic platform with a hole. The line of food paste leads to the capillary attached to the force transducer. (*b*) Habitus of *C. aspersum* (ZMH150005-15). (*c*) Image displaying the area of the radula that is used for feeding; FL, food paste (flour–water mixture), FO, foot, JA, jaw, LI, lip, RA, radula. (*d*–*k*) SEM images of ZMH150005-06: (*d*) view on the broad radula with several tooth types, dashed line boxes show the localities of images (*e,f,j*); (*e*) unworn central tooth with lateral teeth; (*f*) unworn lateral teeth; (*g*) central tooth cusps pictured in a 90° angle for measurements of the contact area; (*h*) worn lateral teeth; (*i*) jaw, wear by abrasion highlighted by black box; (*j*) unworn marginal teeth and (*k*) worn lateral teeth from the side. Scale bars for *b* = 10 mm; *c* = 2 mm; *d,i* = 200 μm; *e,f,g,h,j* = 10 μm; *k* = 20 μm.

Five radulae of *C. aspersum* (table 1) were extracted from freshly killed specimens, dried, mounted on an aluminium stub, coated with carbon and studied using the scanning electron microscope (SEM) LEO 1525 (One Zeiss Drive, Thornwood, NY, USA). The SEM images allowed us an extrapolation of the

potential contact area of teeth with the substrate: all teeth and tooth rows of these five specimens (table 1) were counted. Then, the area of 20 representative tooth cusps (10 lateral and 10 marginal teeth) from the anterior portion of the radula, viewed from above (figure 1g), of each of those five specimens (table 1) was measured with graphical software (see above). Since we knew how much of the radula was used during feeding, we were able to estimate the number of teeth and their total area in contact.

## 2.3. Force measurements

Thirteen individual animals were used in experiments (table 1). They were always weighed before force measurements; most snails were tested more than once (if it was possible), also on different days, depending on the individual behaviour of the snails (table 1). To measure forces generated by the movements of the radula, a horizontally mounted force transducer FORT-10 (World Precision Instruments, Sarasota, FL, USA) with a glass capillary attached to it was used. This equipment and similar set-up was already used for different experiments (e.g. [74]). The capillary was inserted into a 2 mm hole in an acrylic clear platform, and its height could be adjusted. For the measurement of the forces exerted either by horizontal or vertical movement of the snail's feeding apparatus, the force transducer was installed correspondingly. The force transducer was connected to an amplifier (Biopac System, Inc., CA, USA) and computer-based data acquisition and processing system (Acq Knowledge™, Biopac Systems, Inc., v.3.7.0.0, World Precision Instruments, Sarasota, FL, USA).

To stimulate feeding, a sticky food paste composed of wheat flour and water or carrot juice was prepared. A 'runway' made of the paste was deposited on the platform as a line passing over the capillary, which was also coated by the paste. Some paste was applied behind the capillary to 'persuade' the snail to continue feeding beyond it. By using video footage and by registration of a scratching sound, the force peak was assigned to a radula contact with the capillary (see electronic supplementary material, Movie S1). All such events were accompanied by the force peaks of a particular shape (figure 2). The maximum values of these force peaks were determined using Acq Knowledge™ (Biopac Systems, Inc., v.3.7.0.0). Statistical analysis testing for correlations between forces exerted in different directions by the same snail (table 1) was performed using a non-parametric Spearman in R-software (v.R-3.5.2). Since this test needs more than five values per direction, just one single snail ZMH 150005-2 was evaluated (electronic supplementary material, table S1). Analysis of correlations between maximal forces in each direction and mean weight per snail was performed using linear regression with a parametric Pearson correlation test in Excel 2013 (electronic supplementary material, figure S1).

## 2.4. Material properties estimation

Hardness and elastic modulus (Young's modulus) of the teeth were measured by dynamic nanoindentation, a technique which is based on the oscillatory deformation of the smooth test material using a sharp and hard indenter tip [75,76] and has been previously used for characterization of different biological materials (e.g. [77–79]). Two radulae (table 1) were extracted, cleaned and embedded in epoxy (RECKLI®Epoxi WST, Young's modulus of epoxy: 1 GPa). Epoxy was polished with gradual diamond pastes (Buehler MetaDi Ultra Paste 6 µm, 3 µm, 1 µm, Uzwil, Switzerland) and finally fine-polished with a polishing machine (Buehler MataServ 250, Uzwil, Switzerland) by 250 rpm using 0.04 µm suspension (Struers OP-U, Hannover, Germany) to obtain a plain surface displaying the longitudinal section of the teeth for nanoindentation. The dynamic nanoindentation was performed in continuous stiffness measurement mode with the nanoindenter SA2 (MTS Nano Instrument, Oak Ridge, TN, USA) using Berkovich diamond tip (three-sided pyramid) at one location per tooth cusp for 75 mature, fully mineralized marginal and lateral teeth of two animal's wear zone (34 teeth of ZMH 150005-7; 41 of ZMH 150005-8).

In order to gather information about the chemical characterization of the teeth, the five radulae (table 1) used for SEM images were also analysed with the Octane Elect energy-dispersive X-ray spectroscopy (EDS) System with Silicon Drift Detector (energy dispersive X-ray analysis (EDAX), AMETEK Materials Analysis Division, Weiterstadt, Germany) on the SEM LEO 1525 (One Zeiss Drive, Thornwood, NY). Ten teeth of each radula from the anterior part were used.

# 3. Results

## 3.1. Radula, movement and contact area

The radula of *C. aspersum* is isodont with about 140–150 tooth rows (figure 1d) each containing one symmetrical, central tooth (figure 1e) overall flanked by about 60–70 lateral teeth (figure 1f) and

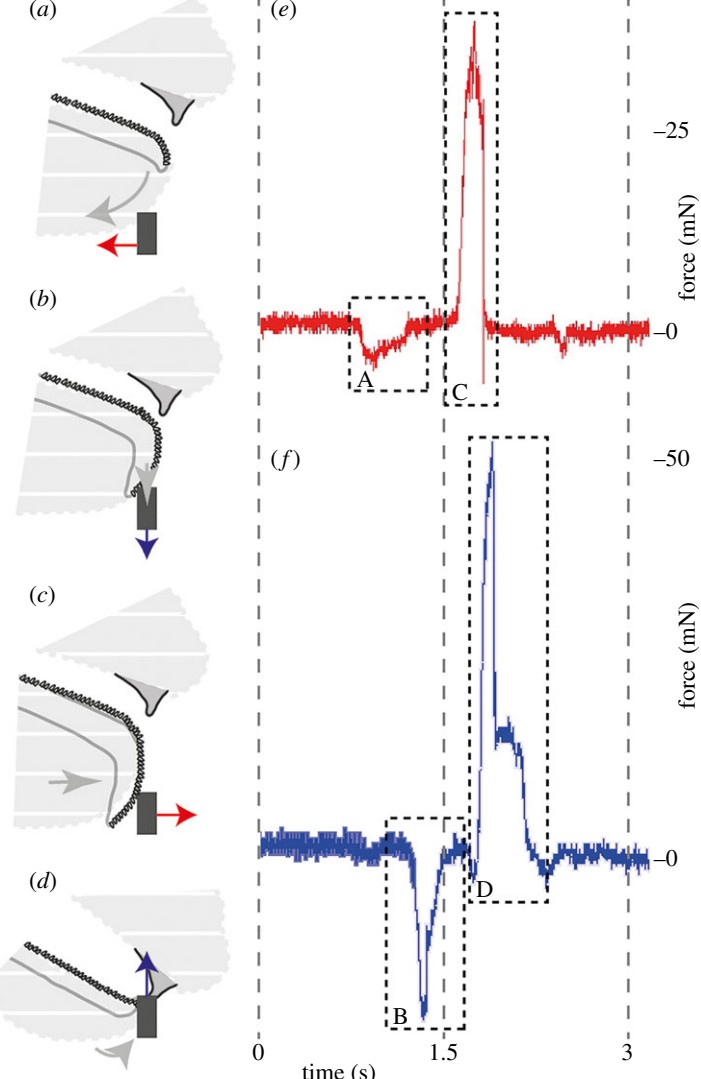

**Figure 2.** Representative force measurements (horizontal, *e*, and vertical, *f*, directions) correlated with the order of the different motions (grey arrows indicate the radula movement); grey dashed area is the snail's body (sagittal view); black box depicts the capillary; the forces exerted on the capillary by the snail are shown as a red arrow for the horizontal direction and as a blue arrow for the vertical direction. (*a*) Force applied by the lip in the posterior horizontal direction, (*b*) force applied by the radula which is pushed on to the substrate (downwards), (*c*) force applied by the radula moving in the horizontal anterior direction and (*d*) force applied by the radula and jaw pulling (upwards).

about 80 marginal teeth (figure 1*g*,*j*). The plain outline of the radula has a surface area of $5.98 \pm 0.24$ mm $\times$ $2.59 \pm 0.13$ mm; the area that is used for feeding is only $2.32 \pm 0.67$ mm$^2$, which is about 15% of the whole area (21–23 tooth rows). This area has a maximum of approximately 3300 teeth in contact with a plain substrate when feeding. The area of the tooth that is in direct contact with a plain surface is the tip (area of central tooth tip $= 0.051 \pm 0.012$ $\mu$m$^2$; the two tips of lateral tooth $=$ each $0.062 \pm 0.015$ $\mu$m$^2$; the four tips of marginal tooth $=$ each $0.0071 \pm 0.0009$ $\mu$m$^2$) resulting in an overall, maximal contact area of about 227 $\mu$m$^2$.

High-speed videos (see electronic supplementary material, Movie S1) of seven snails (table 1) helped us to reveal the radula motion pattern during feeding on (a) lettuce, (b) an acrylic clear surface without food and (c) an acrylic clear surface with a wheat–water food mixture. At the first stage, the lip was extended backwards/posterior opening the mouth (figure 2*a*). At the second stage, the odontophoral cartilage was pulled downwards out of the mouth by radula muscles stretching the radula ribbon until it reached the substrate (figure 2*b*). At the third stage, the stretched radula was pulled anterior rasping the food (figure 2*c*). Finally, the radula moved upwards and snapped the glass capillary with wheat (figure 2*d*) or the lettuce (electronic supplementary material, Movie S1) between the radula and

jaw (figure 1*h*). When snails grazed on an acrylic clear surface without food, they tested the surface with a large part of the radula, but did not extend the whole radula area (the edges were flipped and slightly folded up), and the radula did not interact with the jaw (electronic supplementary material, Movie S1).

## 3.2. Force measurements

Forces exerted by snails during feeding were measured in two different directions (anterior–posterior and upwards–downwards; table 1 for $N$ of experiments and $N$ of force values per snail; figure 4 for the measured forces in each direction for each snail). A typical time dependence of the forces exerted by a snail is presented in figure 2*e,f*. A negative force peak (A) and a positive force peak (C) in figure 2*e* correspond, respectively, to a posterior lip motion (A) and to an anterior radula motion (C). A negative force peak (B) and a positive force peak (D) in figure 2*f* correspond to a downward radula motion (B) and to the food snapping and pulling by the radula and jaw (D), respectively. The highest absolute values of the forces (figure 3) were observed for pushing the radula anterior (avg. $\pm$ s.d. of all: 31.57 $\pm$ 19.23 mN, $n = 143$ values; figure 2*c*) with 106.91 mN in the snail ZMH 150005-9, followed by the pulling upward of snail ZMH 150005-8 with 97.11 mN (avg. $\pm$ s.d. of all: 32.99 $\pm$ 19.82 mN, $n = 115$ values; figure 2*d*). The maximum force measured for the posterior movement of the lip was 55.52 mN measured for snail ZMH 150005-10 (avg. $\pm$ s.d. of all: 19.51 $\pm$ 12.40 mN, $n = 36$ values; figure 2*a*); for the downward movement of the radula 38.48 mN measured in the snail ZMH 150005-8 (avg. $\pm$ s.d. of all: 13.29 $\pm$ 7.66 mN, $n = 67$ values; figure 2*b*). Sometimes, snails repeatedly performed many scratching motions applying increasingly stronger force until reaching the maximum. The maximal forces exerted by snails were independent on their mass (electronic supplementary material, figure S1). However, there were some individual differences: some animals were especially strong during one feeding phase but were not necessarily strong in other feeding phases. In some snails, forces could not be measured at some particular feeding phases (table 1 and figure 4). For $p$-values of Spearman, see electronic supplementary material, table S1.

## 3.3. Material properties estimation

The elasticity modulus and hardness of teeth were determined at an average value of penetration depths in the range of 800–1200 μm with about 50–70 values per individual measurement. The values at lower depth were excluded from the evaluation because of the high variation of parameters due to the surface roughness. In summary, teeth of *C. aspersum* have hardness of 0.48 $\pm$ 0.10 GPa and Young's modulus of 8.97 $\pm$ 1.56 GPa ($n = 75$ measured teeth; ZMH 150005-7: hardness 0.46 $\pm$ 0.08 GPa; ZMH 150005-8: hardness 0.52 $\pm$ 0.06 GPa) and the jaw have hardness of 0.24 $\pm$ 0.11 GPa and Young's modulus of 6.65 $\pm$ 2.4 GPa ($n = 34$). EDAX analyses (see electronic supplementary material, figure S2) revealed O and Ca (0.93 $\pm$ 0.30%) in all teeth and small amounts of Si (0.10–0.16%) in few marginal teeth of one specimen (ZMH 150005-6; see electronic supplementary material, figure S2).

# 4. Discussion

This is the first study revealing radula forces *in vivo* during feeding. The maximal force (107 mN) produced by *C. aspersum* was observed for the radula anterior motion, underlining the importance of this scratching motion. The importance of this grinding and cutting force was also detected in experiments using finite-element analyses on the 3D morphology of radular teeth in other Helicoidea, *Euhadra peliomphala* [80]. The second highest forces were exerted during the upward pulling motion, followed by the posterior lip motion and the radula downward motion. Food particles, rasped off during this latest action, would most probably get lost; therefore, high force at this feeding stage was not expected.

In the radula of *C. aspersum*, teeth do not rely on each transmitting stress as detected in other taxa [46,54], since teeth are situated too far apart (figure 1*k*) and the structures that are in contact with the substrate are the teeth cusps (figure 1*g*). We can estimate that while feeding on a plain surface, the pressure ($p = $ F/A), while scratching (maximal force $=$ 107 mN) applied on the overall surface area of the teeth tips (227 μm$^2$), is 469.871 MPa or 4698.7 bar, which is comparable to some water jet cutters (reaching from 4000 to 6000 bar). If not all teeth are in contact with the substrate, the stress on these structures would be higher (see also [46]). Feeding on a rough or wavy surface would increase mechanical interlocking between teeth and the substrate and would hence increase stress in each tooth resulting in higher risk of fractures or abrasion. In our experiment, the snails were feeding on a sticky,

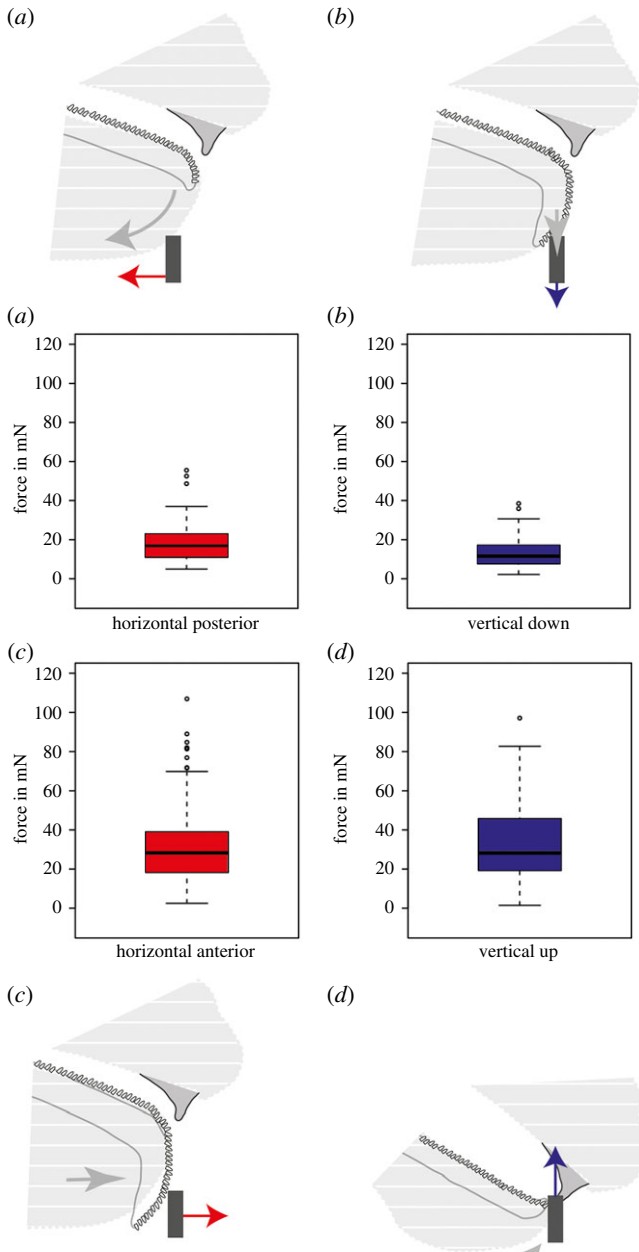

**Figure 3.** General motion pattern (grey arrow indicate the movement) with the summarized force values measured in different directions (forces summarized in boxplots, data from: (a) 10 individual snails with 36 force values; (c) seven individual snails with 67 force values; (c) 12 individual snails with 143 force values; (d) seven individual snails with 115 force values; see also table 1 and figure 4); black box depicts the capillary with the forces (shown as arrows) exerted on it (red = horizontal directions; blue = vertical directions). (a–d) The motion pattern of the feeding apparatus as presented in figure 2: (a) the lip moves posterior horizontally. (b) The radula is moved down in the vertical direction. (c) Anterior motion of radula (scratching). (d) Capillary gripping by the radula and the jaw and pulling it upwards, while rasping off food.

wet paste deposited on a glass capillary, and therefore, it cannot be excluded that the snails may exert stronger forces by eating hard vegetables or cuttlebones.

In general, the snails that were able to exert high forces in one direction were also able to exert higher forces in most other directions (figure 4). But, looking at the results of Spearman, we cannot definitely say that the force exerted in one direction directly influences the forces in the other directions (see electronic supplementary material, table S1), since we do not have data from all studied snails in all possible directions. The one correlation in animal ZMH 150005-14 is probably due to uneven sample sizes.

The material properties of the teeth of *C. aspersum* are comparable to those of wood regarding the hardness and elasticity (see Engineering Tool Box, 2003, https://www.engineeringtoolbox.com/

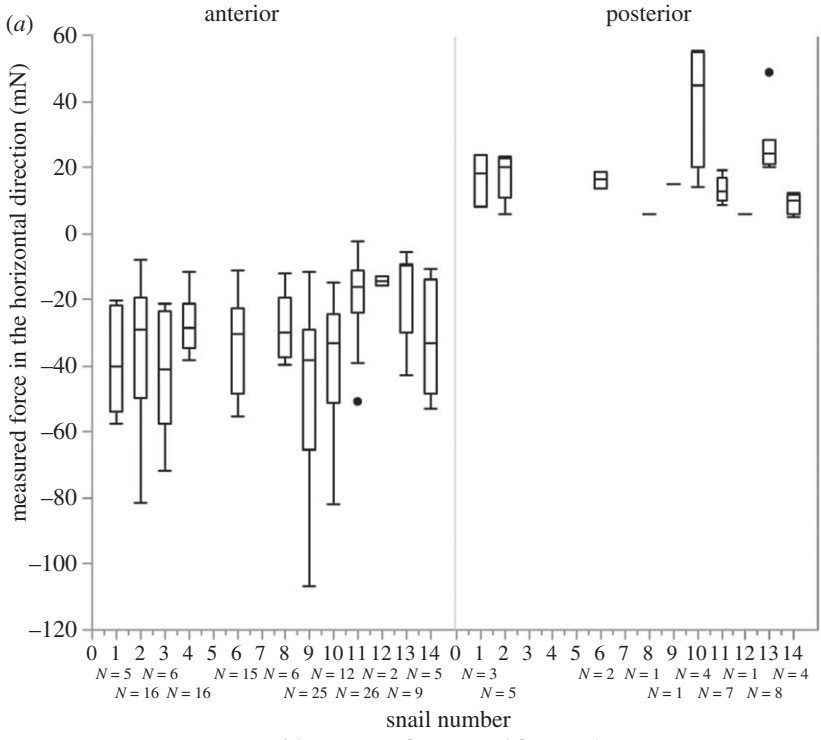

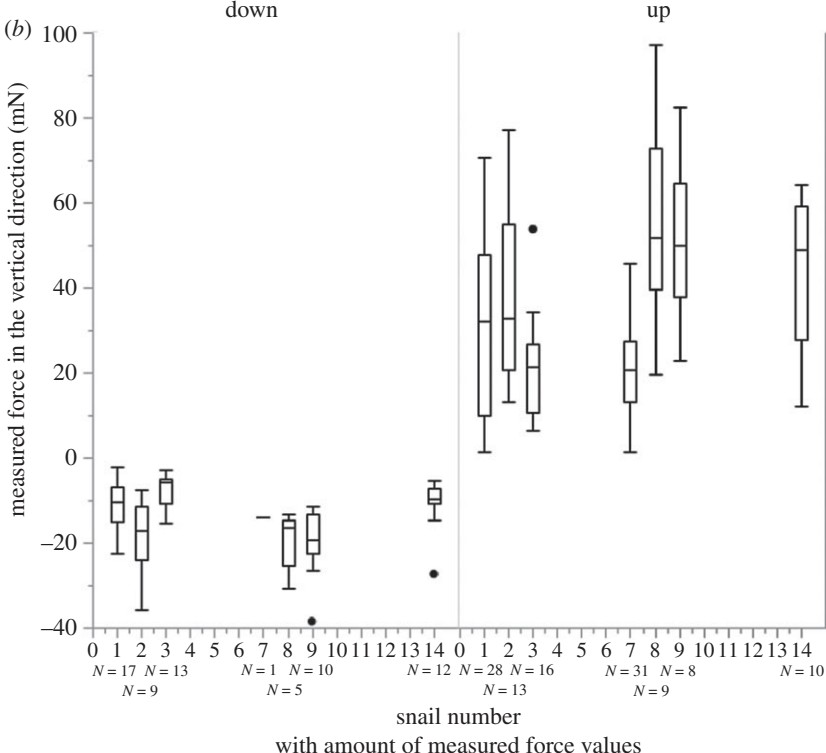

**Figure 4.** Measured forces (mN) for each individual snail in (*a*) horizontal and (*b*) vertical directions. *N* is the amount of force values for each boxplots (see also table 1).

young-modulus-d_417.html [1 May 2018], for values of, e.g. different wood types) but have much lower elasticity modulus (around 9 GPa) in comparison with the previously measured radular teeth of other species of Mollusca, ranging from 16 GPa over 90–125 GPa up to 52–140 GPa [31,35,38,81,82]. The same holds true for the teeth hardness. The previous studies, however, were focused on

Polyplacophora (form an own class within the Mollusca, albeit not closely related to Gastropoda; see [83]) as well as Patellogastropoda (a basal group of the Gastropoda; [83,84]), both phylogenetically distinct from the Heterobranchia *C. aspersum* (see [85] for the phylogenetic position of *C. aspersum* and [84] for the position of Heterobranchia). Also, some Polyplacophora and Patellogastropoda are known for feeding either on the solid substrate or on hardened algae [31,35,38,81,82], but this is probably not the case for *C. aspersum*.

The material properties of the radula of *C. aspersum* had been described by Sollas [86]. In specimens, which were collected during winter time, dry mass consisted to about one-third of Si. Whereas in spring-collected specimens, no Si was found. Here, only small amounts of Si (0.10–0.16%) in only one specimen (ZMH 150005-6) were detected (electronic supplementary material, figure S2). Our experiments were carried out during winter time and the specimens were inventoried in spring, and our specimens were kept in the laboratory and fed with lettuce and carrots. Since only small amounts of calcium (0.93 $\pm$ 0.30%) were found in each tooth, we would consider our studied specimens as marginally mineralized. Many radulae of different species of gastropods and Polyplacophora have been previously analysed and iron, silica or calcium was detected (e.g. [39–42,87–93]), but, as in the present study, for *Cornu*, Mikovari *et al.* [94] found no mineralization in the chitinous radula of *Megathura crenulata*.

In *C. aspersum*, the jaw consists of the material with lower elasticity modulus (6.65 $\pm$ 2.4 GPa) than the one measured for teeth (8.97 $\pm$ 1.56 GPa). Teeth are used for loosening food, so it is plausible that they have to be rather hard, but the jaw is embedded in a supporting tissue and is mainly used as counter bearing to the radula and its sharp edge (figure 1*i*, black box; see electronic supplementary material, Movie S1) for cutting. Since Young's modulus and hardness values usually correlate, the teeth and the jaw possess relatively low hardness (0.48 GPa and 0.24 GPa, respectively), which leads to the lower damage of tissues bearing the teeth, but enhanced teeth abrasion (figure 1*h,i,k*). Interestingly, the teeth of this species are softer than some ingesta, e.g. the cuttlebone (see [95]), but materials can be cut and damaged by using a softer abrasive material and a soft tool that can press the abrasive against surface in concert. The softer tool will simply abrade much faster than the object that is being cut, but since the small contact area of the tooth cusps (227 $\mu m^2$) transmits high local pressure (4698.7 bar) on a plain ingesta surface, the harder material can be cut or pierced with high and gradual abrasion (figure 1*h,k*). In *C. aspersum*, different rates have been reported from 0 to 7 rows d$^{-1}$ depending on age, temperature and hibernation [17,23]. Considering our specimens were mature, not in hibernation and kept at room temperature, we assume a turnover rate of 2.45 rows d$^{-1}$ as in Isarankura & Runham [17]. In chitons (i.e. Polyplacophora), radular teeth have a turnover rate of 0.32–0.36 tooth rows d$^{-1}$ [96,97], but since their teeth are significantly harder [31,35,38,81,82], we would anticipate a weaker wear rate than in *Cornu*'s softer teeth. The connection between not mineralized and hence softer teeth and higher turnover rates is supported by the results of [21] for *Lacuna* with 3 rows d$^{-1}$.

This method for measuring the forces produced by the snail feeding organ *in vivo* can be used in further experiments on gastropods with other radula morphologies and operating at other ecological niches. This potential working direction would be of importance for better understanding functional adaptations of radulae to particular kinds of ingesta or substrate. We anticipate that such a working programme will facilitate new insights, in particular, into trophic specialization potentially involved in gastropod speciation and adaptive radiation.

Data accessibility. See https://figshare.com/s/116b10f1aa1d3ded2d89.

Authors' contributions. S.G. and W.K. conceived this study. T.F. and W.K. conducted the experiments, took care of the animals, analysed the data and worked on the manuscript. W.K. analysed the data, drew the figures and wrote the manuscript. A.K. set up the experiment, supported the data analysis and worked on the manuscript. M.T.N. barcoded the specimen and clarified the nomenclature. M.G. and S.G. contributed to the overall question, worked on the manuscript and provided funding. All authors contributed to the manuscript and approved the final version of the manuscript.

Competing interests. The authors have no competing interests.

Funding. This research received no specific grant from any funding agency in the public, commercial or not-for-profit sectors.

Acknowledgements. The authors thank Peter Stutz (Mineralogisch-Petrographisches Institut, University of Hamburg, Germany) for preparing the samples for nanoindentation, Thomas M. Kaiser (CeNak, University of Hamburg) for his kind support with the video camera and the discussion on the manuscript, Renate Walter (Department of Biology, University of Hamburg) for her support with the SEM imaging and Benedikt Wiggering (CeNak, University of Hamburg) for his input and fruitful discussions. The authors thank the anonymous reviewers for their constructive suggestions.

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
