## [Reviewer comments · Royal Society Open Science]

Review History

RSOS-190222.R0 (Original submission)

Review form: Reviewer 1

Is the manuscript scientifically sound in its present form?

Yes

Are the interpretations and conclusions justified by the results?

No

Is the language acceptable?

No

Is it clear how to access all supporting data?

No

Do you have any ethical concerns with this paper?

No

Have you any concerns about statistical analyses in this paper?

Yes

Recommendation?

Major revision is needed (please make suggestions in comments)

Comments to the Author(s)

This is a revision of a previous submission.

Many of the comments for the previous review have been addressed, but not all.

This paper describes the radula of the land snail *Cornu aspersum*, and several different aspects of the material and mechanical properties of the radula of this species as well as direct measures of the forces applied by the radula when feeding.

The authors have added some information about the radula and work that has been done on the form and function of the radula. But, the authors continue to call this a model system for looking at the properties of the radula. It is not. First, heterobranchs (including the informal poly and paraphyletic group pulmonates) have a very characteristic radular morphology, tooth size, etc. that is not shared with the rest of gastropods, let alone the rest of the mollusca. It is extremely different than most major taxa of gastropods. The heterobranchs are a derived group of gastropods, and within the heterobranchs, terrestrial taxa tend to be derived. To date we do not know if the mineralization and material properties of this species is shared with other heterobranchs. So, it is a case study, and a good starting point for people to look at other species in this group and then start comparing with other taxa. But, it is Not a model system for the gastropod radula.

The inclusion of the table explaining which animals were used for each analysis really improves the clarity of what exactly was done and some of the sample sizes referred to in the text. However, multiple measures from the same individual are not independent. So in most cases, the data as reported have extremely inflated sample sizes. The independent sample sizes are the numbers of different individuals measured. Given that- it would be good to know how much intraindividual variation there was for the different traits and metrics measured, and how that contrasts with the interindividual measures. All data and graphs reported confound these two, making it impossible to really understand what the data are. Given the small independent sample sizes it may not be possible to do statistical analyses on the data. But, that is OK. There is nothing wrong with making this a descriptive paper. We need such information before we can start to compare among species and determine if there are real trends or differences among species, among forms of the radula and among major groups of molluscs. But, we must have clear information about how much of the variation is within versus between individuals.

EDAX was used to determine the mineral content of the radula, confirming older work (Sollas 1907) that *Cornu aspersum* has teeth that are mineralized with silica and calcium. It is not clear if there was interindividual variation in mineralization as was found by Sollas. If not, fine. But, this should be discussed. It would also be useful to compare the amount of silica found in these teeth with other species of mollusc where mineralization has been quantified - particularly in the Patellogastropoda, and if there are examples where it has not been found. For example, Mikovari et al. (2015, J Shellfish Research) found that the keyhole limpet, *Megathura crenulata*, a vetigastropod, does not have minerals in its radula. Many chitons have iron and calcium, but not

silica. Modern electronic databases now make it relatively easy to search much of the literature to find this information. It would really help put this work in a bigger context.

The video clips in the supplementary information are very helpful. However, some the description of radular movement cannot be seen in these clips. And, I do not see the difference in how far the radula is extended from the mouth when food is available and when it is not. Similarly interactions between the radula and the jaw cannot be seen, but are inferred. Quantifying distances or areas might be more convincing. But, inferences are just that. Not facts.

I was hoping that the writing would be more idiomatic with the revision, but not so. So I am including some suggested edits to help with this as well as for clarity. This is not comprehensive, but should help the authors see what is needed.

Specific Comments:

Line 37 - rephrase - species of Mollusca

Line 38 - delete adapted to ingesta - replace with primarily used for feeding

Line 40 - properties of the radular teeth of *Cornu aspersum*

Line 41 - Delete model species

Line 46 - elasticity in this species

Line 47 - replace ingesta with feeding

Line 50 - replace ingesta with feeding and end the sentence there

Line 51 - Delete

Line 54 - recent species in the Class Gastropoda

Line 55 - Taylor and Lewis is a popular book intended for the general public. Please use a scientific reference from the primary literature - there are many to choose from.

Line 60 - Brusca and Brusca is a text book that translates information for students. Use primary references. There are a lot to choose from.

Line 78 - replace does not only comprise to not only includes

Line 81 - chitinous and sometimes mineralized teeth. Radular teeth

Line 84 - radulae have been categorized by the number, type and arrangement of teeth

Line 93 - To date, the vast majority

Line 95 - iron based biominerals and silica

Line 103 - understudied

Lines 107 - 114 - this is taxon specific and must be rewritten. Many if not most taxa do not feed in this way

Line 118 - Feeding can include grazing on soft substrates, collecting microalgae,

Line 121 - to the radula may be used (Markel, 1957). For example,

Line 131 - We used the Mediterranean land snail....., previously known

Line 134 - delete as a model species === it is not a model species. You cannot generalize from this species to most gastropods let alone most molluscs, and, at present, we do not know if other heterobranchs are similar or not.

Line 186 - write out 13 -"thirteen".

Lines 206-207 - performed using a non-parametric Spearman test...

As an aside - because only a single snail could be evaluated - strong inference cannot be drawn

Lines 208-210 - are correlations from a parametric Pearson correlation test? Please clarify

Line 230 - replace 10 with Ten

Line 238 - delete large

Line 239 - of the whole area..... This area has a maximum of approximately 3300 teeth in contact with a substrate when feeding.

Lines 235 - 243 - sample sizes are needed, and it must be clear when numbers come from multiple measures of the same animal and from averages of multiple animals

Line 245 - replace glass with clear

Line 251 - delete we were have seen that

Line 252 - delete just

Lines 255 - 274 - the sample sizes are extremely inflated. Independent samples are different snails, not different observations within a snail. So, ranges within snails can be given. Snail means would be needed to make statements about means for the species. But, both of these types of information are valuable. So reporting both is useful. It just cannot be misleading! Similarly, all graphs need to be changed so that they reflect real sample sizes and real variance.

Line 287 - replace experiment with study

Lines 291-295 - The comparison with the cockroach seems out of place. It would be expected that the mandibles of a cockroach would produce greater force than a snail radula - it is Much larger. It does not seem to add to the discussion, unless other feeding forces have been measured and a range of examples could be given.

Lines 327-335 - Most work on mineral content of the radula has been studied in the Class Polyplacophora and in the Class Gastropoda Patellogastropoda because they are known to mineralize their radula. It is incorrect to assume that they all eat harder food than Cornu. That is clearly not the case. Some do, some do not. I would be surprised if most chitons feed on mineralized algae - certainly the majority of species that I know do not. Many eat fleshy algae and microalgae, even though their teeth are mineralized. The same is true for true limpets. In addition, many terrestrial plants have silica in their cell walls as a defense against herbivores, or are heavily lignified. It is unclear what the natural range of food is for Cornu. And it is unclear if relatives of Cornu also mineralize their teeth. This paragraph is thus misleading and not based on the full range of information known for any of these taxa.

Line 343 - This new method to measure the forces produced by the radula..."

Lines 348-351 - radular replacement rates have been measured for a wide range of species, both that mineralize their radula and those that do not. Comparing Cornu to chitons is like comparing apples and oranges. Better comparisons would be with other gastropods, both those that mineralize their teeth and those that do not. BUT, because there are No Data for Cornu all of this is pure speculation not based on fact. One would expect the rate of replacement to reflect the dynamic balance between the amount of time it takes to make a row of teeth and how fast the teeth wear down when feeding. This has been published. If the authors have information that suggests how fast the teeth would wear and become less useable, then they could estimate a likely production rate to match that. But, it is all speculation, and comparisons with chitons makes no sense.

All data figures need to be corrected to reflect real sample sizes and range of data separating within individual variation from between individual variation.

Decision letter (RSOS-190222.R0)

07-Mar-2019

Dear Ms Krings,

The editors assigned to your paper ("In slow motion: Radula motion pattern and forces exerted to the substrate in the land snail *Cornu aspersum* during feeding") have now received comments from reviewers. We would like you to revise your paper in accordance with the referee and Associate Editor suggestions which can be found below (not including confidential reports to the Editor). Please note this decision does not guarantee eventual acceptance.

Please submit a copy of your revised paper before 30-Mar-2019. Please note that the revision deadline will expire at 00.00am on this date. If we do not hear from you within this time then it will be assumed that the paper has been withdrawn. In exceptional circumstances, extensions may be possible if agreed with the Editorial Office in advance. We do not allow multiple rounds of revision so we urge you to make every effort to fully address all of the comments at this stage. If deemed necessary by the Editors, your manuscript will be sent back to one or more of the original reviewers for assessment. If the original reviewers are not available, we may invite new reviewers.

- Data accessibility

<http://datadryad.org/submit?journalID=RSOS&manu=RSOS-190222>

- Competing interests

- Authors' contributions

All submissions, other than those with a single author, must include an Authors' Contributions section which individually lists the specific contribution of each author. The list of Authors

should meet all of the following criteria; 1) substantial contributions to conception and design, or acquisition of data, or analysis and interpretation of data; 2) drafting the article or revising it critically for important intellectual content; and 3) final approval of the version to be published.

- Acknowledgements

- Funding statement

on behalf of Dr Jake Socha (Associate Editor) and Kevin Padian (Subject Editor)
openscience@royalsociety.org

Associate Editor's comments (Dr Jake Socha):

To fill you in on the process, I determined that the transferred revision needed to be seen by one of the previous reviewers (from the JRSI review). The reviewer has an overall positive impression of the manuscript, deeming the data and study interesting and providing a useful characterization for the field. However, there are still a number of issues that need to be addressed if we are to accept the manuscript for publication. The major items concern the statistical treatment of individuals and the framing of the study as a model system.

Comments to Author:

Reviewers' Comments to Author:

Reviewer: 1

Comments to the Author(s)

This is a revision of a previous submission.

Many of the comments for the previous review have been addressed, but not all.

This paper describes the radula of the land snail *Cornu aspersum*, and several different aspects of the material and mechanical properties of the radula of this species as well as direct measures of the forces applied by the radula when feeding.

The authors have added some information about the radula and work that has been done on the form and function of the radula. But, the authors continue to call this a model system for looking at the properties of the radula. It is not. First, heterobranchs (including the informal poly and paraphyletic group pulmonates) have a very characteristic radular morphology, tooth size, etc. that is not shared with the rest of gastropods, let alone the rest of the mollusca. It is extremely different than most major taxa of gastropods. The heterobranchs are a derived group of gastropods, and within the heterobranchs, terrestrial taxa tend to be derived. To date we do not know if the mineralization and material properties of this species is shared with other heterobranchs. So, it is a case study, and a good starting point for people to look at other species in this group and then start comparing with other taxa. But, it is Not a model system for the gastropod radula.

The inclusion of the table explaining which animals were used for each analysis really improves the clarity of what exactly was done and some of the sample sizes referred to in the text. However, multiple measures from the same individual are not independent. So in most cases, the data as reported have extremely inflated sample sizes. The independent sample sizes are the numbers of different individuals measured. Given that- it would be good to know how much intraindividual variation there was for the different traits and metrics measured, and how that contrasts with the interindividual measures. All data and graphs reported confound these two, making it impossible to really understand what the data are. Given the small independent sample sizes it may not be possible to do statistical analyses on the data. But, that is OK. There is nothing wrong with making this a descriptive paper. We need such information before we can start to compare among species and determine if there are real trends or differences among species, among forms of the radula and among major groups of molluscs. But, we must have clear information about how much of the variation is within versus between individuals.

EDAX was used to determine the mineral content of the radula, confirming older work (Sollas 1907) that *Cornu aspersum* has teeth that are mineralized with silica and calcium. It is not clear if there was interindividual variation in mineralization as was found by Sollas. If not, fine. But, this should be discussed. It would also be useful to compare the amount of silica found in these teeth with other species of mollusc where mineralization has been quantified - particularly in the Patellogastropoda, and if there are examples where it has not been found. For example, Mikovari et al. (2015, J Shellfish Research) found that the keyhole limpet, *Megathura crenulata*, a vetigastropod, does not have minerals in its radula. Many chitons have iron and calcium, but not silica. Modern electronic databases now make it relatively easy to search much of the literature to find this information. It would really help put this work in a bigger context.

The video clips in the supplementary information are very helpful. However, some the description of radular movement cannot be seen in these clips. And, I do not see the difference in how far the radula is extended from the mouth when food is available and when it is not. Similarly interactions between the radula and the jaw cannot be seen, but are inferred. Quantifying distances or areas might be more convincing. But, inferences are just that. Not facts.

I was hoping that the writing would be more idiomatic with the revision, but not so. So I am including some suggested edits to help with this as well as for clarity. This is not comprehensive, but should help the authors see what is needed.

Specific Comments:

Line 37 - rephrase - species of Mollusca

Line 38 - delete adapted to ingesta - replace with primarily used for feeding

Line 40 - properties of the radular teeth of *Cornu aspersum*

Line 41 - Delete model species

Line 46 - elasticity in this species

Line 47 - replace ingesta with feeding

Line 50 - replace ingesta with feeding and end the sentence there

Line 51 - Delete

Line 54 - recent species in the Class Gastropoda

Line 55 - Taylor and Lewis is a popular book intended for the general public. Please use a scientific reference from the primary literature - there are many to choose from.

Line 60 - Brusca and Brusca is a text book that translates information for students. Use primary references. There are a lot to choose from.

Line 78 - replace does not only comprise to not only includes

Line 81 - chitinous and sometimes mineralized teeth. Radular teeth

Line 84 - radulae have been categorized by the number, type and arrangement of teeth

Line 93 - To date, the vast majority

Line 95 - iron based biominerals and silica

Line 103 - understudied

Lines 107 - 114 - this is taxon specific and must be rewritten. Many if not most taxa do not feed in this way

Line 118 - Feeding can include grazing on soft substrates, collecting microalgae,

Line 121 - to the radula may be used (Markel, 1957). For example,

Line 131 - We used the Mediterranean land snail....., previously known

Line 134 - delete as a model species === it is not a model species. You cannot generalize from this species to most gastropods let alone most molluscs, and, at present, we do not know if other heterobranchs are similar or not.

Line 186 - write out 13 - "thirteen".

Lines 206-207 - performed using a non-parametric Spearman test...

As an aside - because only a single snail could be evaluated - strong inference cannot be drawn

Lines 208-210 - are correlations from a parametric Pearson correlation test? Please clarify

Line 230 - replace 10 with Ten

Line 238 - delete large

Line 239 - of the whole area..... This area has a maximum of approximately 3300 teeth in contact with a substrate when feeding.

Lines 235 - 243 - sample sizes are needed, and it must be clear when numbers come from multiple measures of the same animal and from averages of multiple animals

Line 245 - replace glass with clear

Line 251 - delete we were have seen that

Line 252 - delete just

Lines 255 - 274 - the sample sizes are extremely inflated. Independent samples are different snails, not different observations within a snail. So, ranges within snails can be given. Snail means would be needed to make statements about means for the species. But, both of these types of information are valuable. So reporting both is useful. It just cannot be misleading! Similarly, all graphs need to be changed so that they reflect real sample sizes and real variance.

Line 287 - replace experiment with study

Lines 291-295 - The comparison with the cockroach seems out of place. It would be expected that the mandibles of a cockroach would produce greater force than a snail radula - it is Much larger.

It does not seem to add to the discussion, unless other feeding forces have been measured and a range of examples could be given.

Lines 327-335 - Most work on mineral content of the radula has been studied in the Class Polyplacophora and in the Class Gastropoda Patellogastropoda because they are known to mineralize their radula. It is incorrect to assume that they all eat harder food than Cornu. That is clearly not the case. Some do, some do not. I would be surprised if most chitons feed on mineralized algae - certainly the majority of species that I know do not. Many eat fleshy algae and microalgae, even though their teeth are mineralized. The same is true for true limpets. In addition, many terrestrial plants have silica in their cell walls as a defense against herbivores, or are heavily lignified. It is unclear what the natural range of food is for Cornu. And it is unclear if relatives of Cornu also mineralize their teeth. This paragraph is thus misleading and not based on the full range of information known for any of these taxa.

Line 343 - This new method to measure the forces produced by the radula..."

Lines 348-351 - radular replacement rates have been measured for a wide range of species, both that mineralize their radula and those that do not. Comparing Cornu to chitons is like comparing apples and oranges. Better comparisons would be with other gastropods, both those that mineralize their teeth and those that do not. BUT, because there are No Data for Cornu all of this is pure speculation not based on fact. One would expect the rate of replacement to reflect the dynamic balance between the amount of time it takes to make a row of teeth and how fast the teeth wear down when feeding. This has been published. If the authors have information that suggests how fast the teeth would wear and become less useable, then they could estimate a likely production rate to match that. But, it is all speculation, and comparisons with chitons makes no sense.

All data figures need to be corrected to reflect real sample sizes and range of data separating within individual variation from between individual variation.

Author's Response to Decision Letter for (RSOS-190222.R0)

See Appendix A.

RSOS-190222.R1 (Revision)

Review form: Reviewer 1

Is the manuscript scientifically sound in its present form?

Yes

Are the interpretations and conclusions justified by the results?

Yes

Is the language acceptable?

No

Is it clear how to access all supporting data?

Yes

Do you have any ethical concerns with this paper?

No

Have you any concerns about statistical analyses in this paper?

No

Recommendation?

Accept with minor revision (please list in comments)

Comments to the Author(s)

This ms is now much improved in terms of the clarity, particularly regarding sample sizes, and repeated measures on the same animals.

There are a few places that still require more clarity. I have also made comments where the text could be improved.

Line 45 - not clear what is being stated. This sentence needs rewording

Line 48-49 pierced by abrasion

Line 59 - detritus feeders, predators, scavengers

Line 63 - delete already

Line 68 - delete group

Line 69 - 71 - value of radula morphology depends on the group. Some closely related...

Line 77 - includes not only the radula

Line 79 - embedded with transverse and longitudinal rows of teeth that are sometimes mineralized.

Line 92 - needs to be rewritten, is confusing

Line 99-100 - needs to be rewritten - not grammatically correct

Line 101 - radular teeth note, radula = singular, radulae - plural, radular = adj.

Line 107 - replace "pulmonates" with heterobranchs

Line 132 - at a few localities

Lines 222-223 - make clear that this was a total number of teeth from two animals

Line 277 - hardness of teeth were

Line 320 Class Polyplacophora as well as gastropods in the Patellogastropoda

Line 332 delete text after carrots. It is not known if these animals must get new silica from their diet, if they have stores, or if they can recycle it from shed teeth. So, you do not know. All that is known is that they were fed carrots and lettuce and cuttle bones.

Lines 352 - 358 - Radula production/replacement rates have been measured on a wide range of species, including many species without mineralize teeth. Papers in this ms are in the references that give these rates. Isarankura and Runham 1968 - a number of species

Runham and Isarankura also give a rate for *Helix* - *Malacologia* Volume:5 Issue:1

Pages:73 Published: 1966

Padilla et al 1996 *J Moll Stud* 62: 275-280 provide rates for two species of *Lacuna*, which do not have minerals. The authors need to do a better job here of including references and looking at existing information on this

Decision letter (RSOS-190222.R1)

14-May-2019

Dear Ms Krings:

On behalf of the Editors, I am pleased to inform you that your Manuscript RSOS-190222.R1 entitled "In slow motion:

Radula motion pattern and forces exerted to the substrate in the land snail *Cornu aspersum* during feeding" has been accepted for publication in Royal Society Open Science subject to minor revision in accordance with the referee suggestions. Please find the referees' comments at the end of this email.

The reviewers and Subject Editor have recommended publication, but also suggest some minor revisions to your manuscript. Therefore, I invite you to respond to the comments and revise your manuscript.

- Ethics statement

- Data accessibility

<http://datadryad.org/submit?journalID=RSOS&manu=RSOS-190222.R1>

- Competing interests

- Authors' contributions

- Acknowledgements

- Funding statement

Because the schedule for publication is very tight, it is a condition of publication that you submit the revised version of your manuscript before 23-May-2019. Please note that the revision deadline will expire at 00.00am on this date. If you do not think you will be able to meet this date please let me know immediately.

Supplementary files will be published alongside the paper on the journal website and posted on

the online figshare repository (<https://figshare.com>). The heading and legend provided for each supplementary file during the submission process will be used to create the figshare page, so please ensure these are accurate and informative so that your files can be found in searches. Files on figshare will be made available approximately one week before the accompanying article so that the supplementary material can be attributed a unique DOI.

on behalf of Dr Jake Socha (Associate Editor) and Kevin Padian (Subject Editor)
openscience@royalsociety.org

Associate Editor Comments to Author (Dr Jake Socha):

Associate Editor: 1

Comments to the Author:

The major concerns have all been address in this revision. However, a number of small items from reviewer 1 remain to be tackled before final publication.

Reviewer comments to Author:

Reviewer: 1

Comments to the Author(s)

This ms is now much improved in terms of the clarity, particularly regarding sample sizes, and repeated measures on the same animals.

There are a few places that still require more clarity. I have also made comments where the text could be improved.

Line 45 - not clear what is being stated. This sentence needs rewording

Line48-49 pierced by abrasion

Line 59 - detritus feeders, predators, scavengers

Line 63 - delete already

Line 68 - delete group

Line 69 - 71 - value of radula morphology depends on the group. Some closely related...

Line 77 - includes not only the radula

Line 79 - embedded with transverse and longitudinal rows of teeth that are sometimes mineralized.

Line 92 - needs to be rewritten, is confusing

Line 99-100 - needs to be rewritten - not grammatically correct

Line 101 - radular teeth note, radula = singular, radulae - plural, radular = adj.

Line 107 - replace "pulmonates" with heterobranchs

Line 132 - at a few localities

Lines 222-223 - make clear that this was a total number of teeth from two animals

Line 277 - hardness of teeth were

Line 320 Class Polyplacophora as well as gastropods in the Patellogastropoda

Line 332 delete text after carrots. It is not known if these animals must get new silica from their diet, if they have stores, or if they can recycle it from shed teeth. So, you do not know. All that is known is that they were fed carrots and lettuce and cuttle bones.

Lines 352 - 358 - Radula production/replacement rates have been measured on a wide range of species, including many species without mineralize teeth. Papers in this ms are in the references that give these rates. Isarankura and Runham 1968 - a number of species

Runham and Isarankura also give a rate for *Helix* - Malacologia Volume:5 Issue:1

Pages:73 Published: 1966

Padilla et al 1996 J Moll Stud 62: 275-280 provide rates for two species of *Lacuna*, which do not have minerals. The authors need to do a better job here of including references and looking at existing information on this

Author's Response to Decision Letter for (RSOS-190222.R1)

See Appendices B & C.

Decision letter (RSOS-190222.R2)

03-Jun-2019

Dear Ms Krings,

I am pleased to inform you that your manuscript entitled "In slow motion: Radula motion pattern and forces exerted to the substrate in the land snail *Cornu aspersum* during feeding" is now accepted for publication in Royal Society Open Science.

Kind regards,

Andrew Dunn

on behalf of Dr Jake Socha (Associate Editor) and Kevin Padian (Subject Editor)
openscience@royalsociety.org

Appendix A

Reviewers' comments are marked in blue, our responses are in black. All lines given refer to the latest word document with all changes revealed.

Comments to Author:

Reviewers' Comments to Author:

Reviewer: 1

Comments to the Author(s)

This is a revision of a previous submission.

Many of the comments for the previous review have been addressed, but not all.

This paper describes the radula of the land snail *Cornu aspersum*, and several different aspects of the material and mechanical properties of the radula of this species as well as direct measures of the forces applied by the radula when feeding.

The authors have added some information about the radula and work that has been done on the form and function of the radula. But, the authors continue to call this a model system for looking at the properties of the radula. It is not. First, heterobranchs (including the informal poly and paraphyletic group pulmonates) have a very characteristic radular morphology, tooth size, etc. that is not shared with the rest of gastropods, let alone the rest of the mollusca. It is extremely different than most major taxa of gastropods. The heterobranchs are a derived group of gastropods, and within the heterobranchs, terrestrial taxa tend to be derived. To date we do not know if the mineralization and material properties of this species is shared with other heterobranchs. So, it is a case study, and a good starting point for people to look at other species in this group and then start comparing with other taxa. But, it is Not a model system for the gastropod radula.

We are aware of the characteristic radular morphology of heterobranchs. We removed the term “model” and instead state that this is a case study (line 131).

The inclusion of the table explaining which animals were used for each analysis really improves the clarity of what exactly was done and some of the sample sizes referred to in the text. However, multiple measures from the same individual are not independent. So in most cases, the data as reported have extremely inflated sample sizes. The independent sample sizes are the numbers of different individuals measured. Given that- it would be good to know how much intraindividual variation there was for the different traits and metrics measured, and how that contrasts with the interindividual measures. All data and graphs reported confound these two, making it impossible to really understand what the data are. Given the small independent sample sizes it may not be possible to do statistical analyses on the data. But, that is OK. There is nothing wrong with making this a descriptive paper. We need such information before we can start to compare among species and determine if there are real trends or differences among species, among forms of the radula and among major groups of molluscs. But, we must have clear information about how much of the variation is within versus between individuals.

13 individual snails were measured (please see Tab. 1). We conducted 83 single experiments, and several measurements of the same individual were not independent. In Fig. 2 we depict one representative force measurement in one experiment, In Fig. 3 we show the summarized forces in each direction of all experiments. We added another Figure, No. 4, with all forces for each direction for each snail. The exerted forces vary greatly in each snail. Thus, we decided to discuss the general trend between the different feeding motions and the maximum forces (max. forces are also depicted in Supplementary Fig. 1). Sample sizes are added to the description of Fig. 3. In line 307-312 we state that we cannot do correlation.

EDAX was used to determine the mineral content of the radula, confirming older work (Sollas 1907) that *Cornu aspersum* has teeth that are mineralized with silica and calcium. It is not clear if there was interindividual variation in mineralization as was found by Sollas. If not, fine. But, this should be discussed.

We are thankful for this comment, we think that due to our experimental environment, no Si was eaten by our specimens, this is now discussed.

It would also be useful to compare the amount of silica found in these teeth with other species of mollusc where mineralization has been quantified - particularly in the Patellogastropoda, and if there are examples where it has not been found. For example, Mikovari et al. (2015, J Shellfish Research) found that the keyhole limpet, *Megathura crenulata*, a vetigastropod, does not have minerals in its radula. Many chitons have iron and calcium, but not silica. Modern electronic databases now make it relatively easy to search much of the literature to find this information. It would really help put this work in a bigger context.

We agree with the reviewer and are thankful for this comment. We compared the amounts of Si and Ca and would, since the amounts are very small, rather call the radular teeth of our specimens “marginally mineralized”. We included that into our discussion.

The video clips in the supplementary information are very helpful. However, some the description of radular movement cannot be seen in these clips. And, I do not see the difference in how far the radula is extended from the mouth when food is available and when it is not. Similarly interactions between the radula and the jaw cannot be seen, but are inferred. Quantifying distances or areas might be more convincing. But, inferences are just that. Not facts.

The different motion patterns of the radula cannot be seen by just looking from one angle, so all patterns can never been seen within one video sequence (the figures in the manuscript show the idealized motion patterns based on the studies of multiple sequences). Interactions between radula and jaw was only detected when the specimens grasped the glass capillary (this cannot be seen in the videos due to the experimental setup) or while feeding lettuce. In the Supplementary movie S1 (3:35-4:06 min) the movement without food can be seen and it looks like the edges are slightly folded up. We tried to clarify this in line 242-252.

I was hoping that the writing would be more idiomatic with the revision, but not so. So I am including some suggested edits to help with this as well as for clarity.

This is not comprehensive, but should help the authors see what is needed.

Specific Comments:

Line 37 - rephrase - species of Mollusca

Changed accordingly.

Line 38 - delete adapted to ingesta - replace with primarily used for feeding

We left the previous version. Ingesta means: food and everything that is taken in while feeding (food, substrate and dirt). So we cannot exchange the word. It is also important for us to highlight, that adaptations on the teeth to the feedings substrate or food had been recognized in previous studies.

Line 40 - properties of the radular teeth of *Cornu aspersum*

Changed accordingly.

Line 41 - Delete model species

Changed accordingly.

Line 46 - elasticity in this species

Changed accordingly.

Line 47 - replace ingesta with feeding

We left the previous version. Please see above.

Line 50 - replace ingesta with feeding and end the sentence there

We left the previous version. Please see above.

Line 51 – Delete

We left the previous version. In our following manuscripts we plan to consider adaptations of radulae to the feeding substrate in the framework of gastropod evolution. So, for us, this point is of high importance.

Line 54 - recent species in the Class Gastropoda

Changed accordingly.

Line 55 - Taylor and Lewis is a popular book intended for the general public. Please use a scientific reference from the primary literature - there are many to choose from.

Line 60 - Brusca and Brusca is a text book that translates information for students. Use primary references. There are a lot to choose from.

Changed accordingly.

Line 78 - replace does not only comprise to not only includes

Changed accordingly.

Line 81 - chitinous and sometimes mineralized teeth. Radular teeth

Radula teeth and radular teeth are equally used in the literature (<https://www.sciencedirect.com/science/article/pii/S1047847715300769>, <https://www.sciencedirect.com/science/article/pii/0047720672900180>).

Line 84 - radulae have been categorized by the number, type and arrangement of teeth

Changed accordingly.

Line 93 - To date, the vast majority

Changed accordingly.

Line 95 - iron based biominerals and silica

Changed accordingly.

Line 103 – understudied

Changed accordingly.

Lines 107 - 114 - this is taxon specific and must be rewritten. Many if not most taxa do not feed in this way

We clarified in the text, Mackenstedt and Märkel (2001) described the movement of some terrestrial “pulmonata”, but unfortunately without naming the species.

Line 118 - Feeding can include grazing on soft substrates, collecting microalgae,

Changed accordingly.

Line 121 - to the radula may be used (Markel, 1957). For example,
Changed accordingly.

Line 131 - We used the Mediterranean land snail....., previously known
Changed accordingly.

Line 134 - delete as a model species === it is not a model species. You cannot generalize from this species to most gastropods let alone most molluscs, and, at present, we do not know if other heterobranchs are similar or not.
Changed accordingly.

Line 186 - write out 13 -“thirteen”.
Changed accordingly.

Lines 206-207 - performed using a non-parametric Spearman test...
As an aside - because only a single snail could be evaluated - strong inference cannot be drawn
Changed accordingly. This is something we know and states in line 311-312.

Lines 208-210 - are correlations from a parametric Pearson correlation test? Please clarify
Changed accordingly.

Line 230 - replace 10 with Ten
Changed accordingly.

Line 238 - delete large
Changed accordingly.

Line 239 - of the whole area..... This area has a maximum of approximately 3300 teeth in contact with a substrate when feeding.
Changed accordingly.

Lines 235 - 243 - sample sizes are needed, and it must be clear when numbers come from multiple measures of the same animal and from averages of multiple animals.
Sample sizes are given in the materials & methods section and in Tab. 1.

Line 245 - replace glass with clear
Changed accordingly.

Line 251 - delete we were have seen that
Changed accordingly.

Line 252 - delete just
Changed accordingly.

Lines 255 - 274 - the sample sizes are extremely inflated. Independent samples are different snails, not different observations within a snail. So, ranges within snails can be given. Snail means would be needed to make statements about means for the species. But, both of these types of information are valuable. So reporting both is useful. It just cannot be misleading! Similarly, all graphs need to be changed so that they reflect real sample sizes and real variance.

13 individual snails were measured (please see Tab. 1) with 83 single experiments. The exerted forces vary greatly in each snail and even within a single experiment (now, we included a new Figure 4 to show all measured forces for each snail in each direction). So, we decided to discuss the maximum forces and the general trend between the different feeding motions. Sample sizes are added to the description of Fig. 3.

Line 287 - replace experiment with study
Changed accordingly.

Lines 291-295 - The comparison with the cockroach seems out of place. It would be expected that the mandibles of a cockroach would produce greater force than a snail radula - it is Much larger. It does not seem to add to the discussion, unless other feeding forces have been measured and a range of examples could be given.

This is true, the comparison with the cockroach was removed.

Lines 327-335 - Most work on mineral content of the radula has been studied in the Class Polyplacophora and in the Class Gastropoda Patellogastropoda because they are known to mineralize their radula. It is incorrect to assume that they all eat harder food than *Cornu*. That is clearly not the case. Some do, some do not. I would be surprised if most chitons feed on mineralized algae - certainly the majority of species that I know do not. Many eat fleshy algae and microalgae, even though their teeth are mineralized. The same is true for true limpets. In addition, many terrestrial plants have silica in their cell walls as a defense against herbivores, or are heavily lignified. It is unclear what the natural range of food is for *Cornu*. And it is unclear if relatives of *Cornu* also mineralize their teeth. This paragraph is thus misleading and not based on the full range of information known for any of these taxa.

For some Polyplacophora and Patellogastropoda we find in literature that they feed on solid substrates or on hardened algae (Weaver et al., 2010; Lu and Barber, 2012; Grunenfelder et al., 2014; Barber et al., 2015; Ukmar-Godec, 2017). We truly do not know the food range of *Cornu* but they usually do not loosen food from a solid surface.

Line 343 - This new method to measure the forces produced by the radula..."
Changed accordingly.

Lines 348-351 - radular replacement rates have been measured for a wide range of species, both that mineralize their radula and those that do not. Comparing *Cornu* to chitons is like comparing apples and oranges. Better comparisons would be with other gastropods, both those that mineralize their teeth and those that do not. BUT, because there are No Data for *Cornu* all of this is pure speculation not based on fact. One would expect the rate of replacement to reflect the dynamic balance between the amounts of time it takes to make a row of teeth and how fast the teeth wear down when feeding. This has been published. If the authors have information that suggests how fast the teeth would wear and become less useable, then they could estimate a likely production rate to match that. But, it is all speculation, and comparisons with chitons makes no sense.

Here we hypothesize how *Cornu* can loosen cuttlebone, even though this is harder than the teeth. From a physician point of view this can be explained with high abrasion. So in this context we would expect *Cornu* to replace teeth faster, but – since we do not have data – these are hypotheses and we state this in the discussion. We compare *Cornu* with Polyplacophora and Patellogastropoda, because hardness and elasticity were previously studied in their radular teeth (Weaver et al., 2010; Lu and Barber, 2012; Grunenfelder et al., 2014; Barber et al., 2015; Ukmar-Godec, 2017) and there is no data for Heterobranchia so far.

All data figures need to be corrected to reflect real sample sizes and range of data separating within individual variation from between individual variation.
Changed accordingly.

Appendix B

Dear Mr. Dunn,

we appreciate the opportunity to submit this manuscript again to the Royal Society Open Science after minor revision and are again very grateful for the input, time and efforts of editors and reviewers which we consider very helpful for improving our paper. We have decided to submit our reworked manuscript, as we feel it is worthwhile to further improve our paper along the lines suggested in the reviews.

We hope that this new version of the manuscript meets expectations and standards of RSOS journal.

Yours sincerely,

Wencke Krings, Taissa Faust, Alexander Kovalev, Marco T. Neiber, Matthias Glaubrecht, and Stanislav Gorb.

Appendix C

Reviewers' comments are marked in blue, our responses are in black. All lines given refer to the latest word document with all changes revealed.

Comments to the Author:

The major concerns have all been address in this revision. However, a number of small items from reviewer 1 remain to be tackled before final publication.

Reviewer comments to Author:

Reviewer: 1

Comments to the Author(s)

This ms is now much improved in terms of the clarity, particularly regarding sample sizes, and repeated measures on the same animals.

There are a few places that still require more clarity. I have also made comments where the text could be improved.

Line 45 - not clear what is being stated. This sentence needs rewording
Changed accordingly.

Line48-49 pierced by abrasion
Changed accordingly.

Line 59 - detritus feeders, predators, scavengers
Changed accordingly.

Line 63 - delete already
Changed accordingly.

Line 68 - delete group
Changed accordingly.

Line 69 - 71 - value of radula morphology depends on the group. Some closely related...
Minor change to avoid repetition.

Line 77 - includes not only the radula
Changed accordingly.

Line 79 - embedded with transverse and longitudinal rows of teeth that are sometimes mineralized.
Changed accordingly.

Line 92 - needs to be rewritten, is confusing
Changed accordingly.

Line 99-100 - needs to be rewritten - not grammatically correct
Changed accordingly.

Line 101 - radular teeth note, radula = singular, radulae - plural, radular = adj.
Changed accordingly.

Line 107 - replace "pulmonates" with heterobranchs

Changed accordingly.

Line 132 - at a few localities

Changed accordingly.

Lines 222-223 - make clear that this was a total number of teeth from two animals

Changed accordingly.

Line 277 - hardness of teeth were

Changed accordingly.

Line 320 Class Polyplacophora as well as gastropods in the Patellogastropoda

It's important for us to clarify the systematic position of the Polyplacophora and Patellogastropoda with all the citing literature and to compare this with *Cornu*. We want to show, that the vast majority of gastropods have not been analysed with nanoindentation and were hence not able to compare our species with more closely related taxa. This is the reason why we left it as previous.

Line 332 delete text after carrots. It is not known if these animals must get new silica from their diet, if they have stores, or if they can recycle it from shed teeth. So, you do not know. All that is known is that they were fed carrots and lettuce and cuttle bones.

Changed accordingly.

Lines 352 - 358 - Radula production/replacement rates have been measured on a wide range of species, including many species without mineralize teeth. Papers in this ms are in the references that give these rates. Isarankura and Runham 1968 - a number of species

Runham and Isarankura also give a rate for *Helix* - Malacologia Volume:5 Issue:1
Pages:73 Published: 1966

Padilla et al 1996 J Moll Stud 62: 275-280 provide rates for two species of *Lacuna*, which do not have minerals. The authors need to do a better job here of including references and looking at existing information on this

We are very thankful for this and included this in our discussion.